# Prevalence and appropriateness of in-person versus not-in-person ambulatory antibiotic prescribing in an integrated academic health system: A cohort study

**Tiffany Brown[1], Ji Young Lee[1], Adriana Guzman[1], Michael A. Fischer[2], Mark W. Friedberg[3,4], Kao-Ping Chua[5], Jeffrey A. Linder[1] ***

**1** Division of General Internal Medicine, Department of Medicine, Northwestern University Feinberg School of Medicine, Chicago, IL, United States of America, **2** Department of Medicine, Section of General Internal Medicine, Boston Medical Center, Boston University Chobanian & Avedisian School of Medicine, Boston, MA, United States of America, **3** Blue Cross Blue Shield of Massachusetts, Boston, MA, United States of America, **4** Division of General Internal Medicine, Department of Medicine, Brigham and Women's Hospital, Boston, MA, United States of America, **5** Susan B. Meister Child Health and Evaluation Research Center, Department of Pediatrics, University of Michigan Medical School, Ann Arbor, MI, United States of America

* jlinder@northwestern.edu

**Data Availability Statement:** Restrictions apply to the availability of these data they are potentially identifiable, sensitive, protected health information

## Abstract

### Objectives

Ambulatory antibiotic stewardship generally aims to address the appropriateness of antibiotics prescribed at in-person visits. The prevalence and appropriateness of antibiotics prescribed outside of in-person visits is poorly studied.

### Design and setting

Retrospective cohort study of all ambulatory antibiotic prescribing in an integrated health delivery system in the United States.

### Participants

Antibiotic prescribers and patients receiving oral antibiotic prescriptions between January 2016 and December 2019.

### Main outcome measures

Proportion of antibiotics prescribed with in-person visits or not-in-person encounters (e.g., telephone, refills). Proportion of prescriptions in in 5 mutually exclusive appropriateness groups: 1) chronic antibiotic use; 2) antibiotic-appropriate; 3) potentially antibiotic-appropriate; 4) non-antibiotic-appropriate; and 5) not associated with a diagnosis.

### Results

Over the 4-year study period, there were 714,057 antibiotic prescriptions ordered for 348,739 unique patients by 2,391 clinicians in 467 clinics. Patients had a mean age of 41

of Northwestern Medicine patients. Making the data externally available requires the approval of the Northwestern Medicine Data Steward and Privacy Official, Rayan Venkatesh, Director of Integrity, Corporate Compliance and Integrity (email, rvenkate@nm.org; phone, 312-695-5913).

**Funding:** This research was supported by grant number R01HS024930 from the Agency for Healthcare Research and Quality (AHRQ). Dr. Linder is also supported by grants from the National Institute on Aging (P30AG059988, R01AG069762, R01AG074245, P30AG024968, R01AG070054, R33AG057395) and the Agency for Healthcare Research and Quality (R01HS026506, R01HS028127). The contents of this product are solely the responsibility of the authors and do not necessarily represent the official views of or imply endorsement by AHRQ or the U.S. Department of Health and Human Services. The funders had no role in study design, data collection or analysis, decision to publish, or preparation of the manuscript.

**Competing interests:** The authors have declared that no competing interests exist.

years old, were 61% female, and 78% White. Clinicians were 58% women; 78% physicians; and were 42% primary care, 39% medical specialists, and 12% surgical specialists. Overall, 81% of antibiotics were prescribed with in-person visits and 19% without in-person visits. The most common not-in-person encounter types were telephone (10%), orders only (5%), and refill encounters (3%). Of all antibiotic prescriptions, 16% were for chronic use, 15% were antibiotic-appropriate, 39% were potentially antibiotic-appropriate, 22% were non-antibiotic-appropriate, and 8% were not associated with a diagnosis. Antibiotics prescribed in not-in-person encounters were more likely to be chronic (20% versus 15%); less likely to be associated with appropriate or potentially appropriate diagnoses (30% versus 59%) or non-antibiotic-appropriate diagnoses (8% versus 25%); and more likely to be associated with no diagnosis (42% versus <1%).

## Conclusions

Ambulatory stewardship interventions that focus only on in-person visits may miss a large proportion of antibiotic prescribing, inappropriate prescribing, and antibiotics prescribed in the absence of any diagnosis.

## Introduction

Antibiotics are over-prescribed in ambulatory care, increase the risk of adverse events, and promote the development of antibiotic-resistant bacteria [1–9]. Antibiotic stewardship interventions aim to reduce inappropriate antibiotic prescribing to improve patient safety and, more broadly, maintain the effectiveness of antibiotics [10, 11]. Many ambulatory antibiotic stewardship studies, measures, and recommendations examine or intervene on antibiotic prescribing at ambulatory in-person visits [10, 12–20].

However, antibiotics can be prescribed without in-person visits, for example when patients and clinicians communicate via telephone call or electronic patient portal messages. Antibiotic prescribing at these "not-in-person encounters" are overlooked in many stewardship initiatives and in research. Not-in-person encounters may also have lower quality because of an inability to do a physical examination. Prior publications have examined the appropriateness of outpatient antibiotic prescribing across all settings using claims data, but these studies did not distinguish between antibiotics prescribed with and without in-person encounters [5, 21–23].

Our objectives were, among all antibiotics prescribed within a large integrated health system using an electronic health record (EHR), to 1) assess the prevalence of not-in-person antibiotic prescriptions and 2) assess the appropriateness of antibiotic prescribing across both in-person and not-in-person encounters, thus providing a comprehensive picture of outpatient antibiotic prescribing [24]. We used EHR data because they have more detailed information on encounter type compared with claims data. Moreover, the indication for antibiotics can more readily be ascertained using EHR data, using problem lists and diagnosis codes.

## Materials and methods

### Study design and setting

We conducted a retrospective cohort study of all ambulatory, oral antibacterial antimicrobial prescriptions ordered in the EHR between January 1, 2016 and December 31, 2019 by

clinicians at Northwestern Medicine, prior to the COVID pandemic and the introduction of synchronous, remote visits. Northwestern Medicine is a large, integrated academic health delivery system with hundreds of sites, about 4000 physicians, and 30,000 staff in the Chicago area (Illinois, USA). Northwestern uses a single EHR (Epic; Verona, WI). Northwestern aggregates clinical data in the Northwestern Medicine Enterprise Data Warehouse (NMEDW) [25]. The Northwestern University Institutional Review Board approved this study with a waiver of informed consent for retrospective review of EHR data. Patients and the public were not involved in the design, conduct, reporting, or dissemination of this work.

## Antibiotics

We used an EHR-based medication grouper to identify oral antibiotic prescriptions (**S1 File**). We included antibiotics prescribed by clinicians of all specialties to patients at an ambulatory clinic, regardless of patient age; whether the patient lived at home, at a care facility, or elsewhere; or any other patient factor. We excluded non-oral antibiotics and methenamine. The unit of analysis is the prescription: individual prescribers could contribute multiple antibiotic prescriptions and individual patients could be associated with multiple antibiotic prescriptions.

## In-person visits or not-in-person encounter types

The EHR organizes units of work into "encounters" of differing types such as clinic visits, telephone calls, electronic patient portal messages, and refills, among others. All prescriptions are generated within a defined encounter type in the EHR. Diagnoses are not required within all encounter types. For example, at least one diagnosis was required for an in-person encounter, but not for a telephone or patient portal encounter.

We classified encounters as either in-person or not-in-person. We considered the following encounter types not-in-person: telephone, patient portal, orders only, refill, and other. Orders only encounters are created when clinicians enter a patient's record solely to enter orders (e.g., prescriptions) without any other activity or documentation. Any prescriptions entered in this manner are, by definition, non-visit-based since a visit would generate a different type of encounter in the electronic record. During the study period, which predated the COVID-19 pandemic, we did not have separate synchronous telephone, synchronous video, or asynchronous e-visit (e.g., questionnaire-based) visit types.

We validated this classification with manual chart review. Two study team members (AG, TB) independently reviewed 54 charts with "other" encounter classification type to determine whether any of these rarely used documentation mechanisms could be classified as in-person visits, but all "other" types were determined to be not-in-person encounters.

## Appropriateness

In the Northwestern EHR, the encounters in which antibiotics are ordered can be associated with zero, one, or more than one diagnosis codes. We previously defined all 94,249 International Classification of Diseases, 10$^{th}$ Edition, Clinical Modification (ICD-10-CM) diagnosis codes as always, sometimes, or never justifying antibiotic prescribing [5]. Using a slight modification of this scheme, we classified all antibiotic prescriptions into 5 mutually exclusive groups: chronic antibiotic use, antibiotic-appropriate, potentially antibiotic-appropriate, non-antibiotic-appropriate, or not associated with any diagnosis (**S1 Data**). Our scheme erred on the side of being lenient to prescribers, increasing the likelihood that an antibiotic prescription would be associated with an antibiotic-appropriate diagnosis (e.g., we would consider the

antibiotic appropriate if there were both antibiotic-appropriate and non-antibiotic-appropriate codes present).

**Chronic antibiotic use.** We identified 690 ICD-10-CM codes that could be associated with a chronic infection or might be indications for chronic antibiotic use (e.g., chronic osteomyelitis, acne, cystic fibrosis, emphysema) [22, 23]. We examined diagnosis codes and problem list diagnoses present in the 6 months prior to antibiotic prescribing and, if present, considered those to be for chronic antibiotic use, regardless of the nature of any other associated diagnosis codes during this lookback period.

**Antibiotic-appropriate, potentially antibiotic-appropriate, non-antibiotic-appropriate.** For the remaining antibiotics, we compiled all encounter diagnoses on the same day as the antibiotic prescription, regardless of whether diagnoses were coded at in-person versus not-in-person encounters, and regardless of which prescriber coded the diagnosis (i.e., the antibiotic could be prescribed and the diagnosis applied by different clinicians on the same day). Additionally, we compiled all diagnosis codes coded by the antibiotic prescriber up to 21 days later, as our manual chart review revealed instances in which a clinician would initiate an encounter, apply a diagnosis, and, within that same encounter, prescribe an antibiotic up to 21 days later. A subsequent analysis with the data from the present study revealed subtle differences in results using different durations between the antibiotic prescription and diagnosis codes [26].

If at least one of the 9,495 "always" diagnosis codes was present, we classified the prescription as antibiotic-appropriate. If at least one of the 11,143 "sometimes" diagnosis codes but no "always" code was present, we classified the prescription as potentially antibiotic-appropriate. If at least one "never" code but no "always" or "sometimes" code was present, we classified the antibiotic prescription as "non-antibiotic-appropriate."

**No diagnosis.** If there was no diagnosis code, we considered the prescription associated with no diagnosis.

Within each category, we examined the 5 most common diagnoses reported in the prior 6 months for chronic antibiotics, and the 5 most common diagnoses reported on the same day as prescriptions classified as antibiotic-appropriate, potentially antibiotic-appropriate and non-antibiotic-appropriate.

## Descriptive variables

Antibiotic prescriptions were the unit of analysis. In examining differences between in-person and not-in person prescriptions, we examined clinic, patient, and clinician characteristics. We defined a clinic as a uniquely named practice site with a unique physical address. We used registration data for patient age, gender, ethnicity, race, marital status, and insurance. We calculated patients' neighborhood household income by matching patient zip codes to 2013–2017 American Community Survey data. We used the EHR to calculate the number of patient comorbidities, prescriptions, clinic visits, emergency department visits, hospitalizations, and whether the patient had a listed primary care physician. For clinicians, we used the listing in the EHR to determine clinician type (e.g., physician, physician assistant, etc.).

We classified physician specialty as primary care, medical specialty, surgical specialty, and other. We calculated the number of years in practice by subtracting the year of medical school graduation from 2019. We calculated ambulatory clinical full-time equivalence by dividing the annual number of half day clinic sessions (before and after 12PM) during which clinicians had at least 3 patient encounters by total half days of employment. Our calculation only included time in ambulatory clinic and did not include inpatient service, operating time, administrative, teaching, research, or other activities. We used the EHR to measure the number of total prescriptions and number of antibiotic prescriptions written during the study period.

## Statistical analysis

We used descriptive statistics to describe the cohort of prescriptions, patients, and prescribing clinicians, to calculate the prevalence of not-in-person antibiotic prescriptions, and to calculate the proportion of not-in-person and in-person prescriptions assigned to each of the 5 mutually exclusive appropriateness categories. Because the large sample size makes clinically insignificant differences statistically significant, the lack of independence among some comparisons, and interest in taking an all-antibiotic prescribing stewardship perspective, we did not perform statistical significance testing, clustered analyses, or adjusted analyses. Rather, we considered absolute differences of 5% or greater to be clinically significant [22, 23]. All analyses were performed using SAS v9.4 (SAS Institute Inc., Cary, NC).

# Results

In the four-year study period from 2016 to 2019, there were 714,057 antibiotic prescriptions ordered for 348,739 unique patients by 2,391 clinicians in 467 clinics. The most common antibiotic classes were penicillins (31%), macrolides (20%), tetracyclines (11%), and cephalosporins (10%). Patients were 61% female with a mean age of 41 (SD±22) years old; 52% had private insurance (**Table 1**). Over half (59%) of prescribing clinicians were physicians, 42% practiced in primary care, and 30% were > 0.5 ambulatory clinical full time equivalent.

## In-person versus not-in-person prescriptions

Although the groups were not independent, patients who received any not-in-person antibiotic prescriptions were more likely to be older, women, married/partnered, have Medicare, take more medication, make more visits, have a primary care physician, and be hospitalized during the study period compared with patients who received any in-person antibiotic prescription (**Table 1**). Seventy three percent of clinicians prescribed both in-person and not-in-person antibiotics during the study period; 27% prescribed only one or the other.

Over the four-year study period, 81% (n = 578,130) antibiotics were prescribed with an in-person encounter and 19% (n = 135,927) with a not-in-person encounter (**Table 2**). The most common types of not-in-person encounters in which antibiotics were prescribed were telephone (10% of all antibiotics prescribed with not-in-person encounters), orders-only (5%), and refill (3%) encounters.

**Antibiotic appropriateness.** The 5 most common diagnoses in each appropriateness category are listed in the footnote of **Table 2**. Across all encounter types, chronic antibiotic use accounted for 16% of all antibiotic prescriptions, while antibiotic-appropriate and potentially antibiotic-appropriate prescriptions collectively accounted for 54% (**Table 2**). Non-antibiotic-appropriate diagnoses accounted for 22% of antibiotic prescriptions. Eight percent of antibiotic prescriptions were associated with no diagnosis. Thus, 30% of antibiotics were either non-antibiotic-appropriate or not associated with any diagnosis.

Antibiotics prescribed in not-in-person encounters were more likely to be chronic (20% versus 15% compared to antibiotics prescribed with in-person visits). Among not-in-person encounter types, antibiotic prescriptions were most likely to be for chronic use among refill (29%) and patient portal (22%) encounters. Not-in-person antibiotics were less likely to be associated with appropriate or potentially appropriate diagnoses (30% versus 59% for in-person visits). Not-in-person antibiotics, depending on the specific encounter type, were variably less likely to be associated with antibiotic-appropriate diagnoses. Not-in-person antibiotics were less likely associated with potentially-antibiotic-appropriate diagnoses (20% versus 43%). Except for the relatively rare "other" not-in-person encounters, not-in-person encounters were less likely to be associated with non-antibiotic appropriate diagnoses (8% versus 25%).

**Table 1. Patient, clinician, and physician characteristics.**

| Patient Characteristics | Total (N = 348,739[a]) | Any In-Person (N = 313,980[b]) | Any Not-In-Person (N = 82,219) |
|---|---|---|---|
| Age in years, mean (SD) | 41 (22) | 40 (22) | 49 (21) |
| Female[c], n (%) | 211,312 (61) | 188,326 (60) | 55,631 (68) |
| Ethnicity–Hispanic or Latino, n (%) | 28,809 (8) | 26,050 (8) | 6,145 (7) |
| Race, n (%) | | | |
| Asian | 14,624 (4) | 13,229 (4) | 2,910 (4) |
| Black | 20,930 (6) | 17,847 (6) | 6,113 (7) |
| White | 272,786 (78) | 247,158 (79) | 64,178 (78) |
| Other/Unknown | 40,399 (12) | 35,746 (11) | 9,018 (11) |
| Marital status, n (%) | | | |
| Married/Partner | 161,032 (46) | 142,430 (45) | 43,250 (53) |
| Divorced/Separated/Widowed | 25,250 (7) | 21,747 (7) | 8,864 (11) |
| Single | 152,086 (44) | 140,738 (45) | 27,727 (34) |
| Other/Unknown | 10,371 (3) | 9,065 (3) | 2,378 (3) |
| Insurance, n (%) | | | |
| Private | 182,473 (52) | 168,427 (54) | 34,754 (42) |
| Medicaid | 22,305 (6) | 20,757 (7) | 3,752 (5) |
| Medicare | 46,967 (14) | 40,159 (13) | 16,921 (21) |
| Self-pay/Other | 96,994 (28) | 84,637 (27) | 26,792 (33) |
| Median neighborhood annual household income by zip code, $ (IQR) | 81,667 (66,281, 100,116) | 84,040 (73,083, 100,994) | 83,145 (66,771, 100,116) |
| Comorbidities, median (IQR) | 0 (0, 1) | 0 (0, 1) | 0 (0, 1) |
| # of other prescriptions, median (IQR) | 3 (1, 5) | 2 (1, 5) | 4 (2, 7) |
| # of visits with any clinicians in period, median (IQR) | 8 (4, 17) | 8 (4, 17) | 13 (6, 24) |
| Counts of in-person visits between patient and prescribing clinician during the study period, mean (SD) / median (IQR) | 3.7 (4.0) / 2 (1, 5) | 3.7 (4.1) / 2 (1, 5) | 4.2 (4.5) / 3 (1, 5) |
| # of emergency department visits in period, n (%) | | | |
| 1 | 332,579 (95) | 300,268 (96) | 76,550 (93) |
| 0 | 11,927 (3) | 10,175 (3) | 3,979 (5) |
| 2+ | 4,233 (1) | 3,537 (1) | 1,690 (2) |
| # of hospitalizations in period, n (%) | | | |
| 0 | 282,950 (81) | 257,360 (82) | 58,650 (71) |
| 1 | 36,708 (11) | 3,917 (10) | 11,933 (15) |
| 2 | 12,631 (4) | 10,875 (3) | 4,705 (6) |
| 3+ | 16,450 (5) | 13,828 (4) | 6,931 (8) |
| Primary care clinician listed, n (%) | 292,336 (84) | 262,603 (84) | 74,059 (90) |
| **Clinician Characteristics** | **N = 2,391** | **N = 2,175[d]** | **N = 1,959** |
| Female[e], n (%) | 1,383 (58) | 1,279 (59) | 1,168 (60) |
| Clinician type, n (%) | | | |
| Physician Assistant | 198 (8) | 188 (9) | 165 (8) |
| APN/NP/Midwife[f] | 311 (13) | 275 (13) | 277 (14) |
| Resident/Fellow | 463 (19) | 428 (20) | 226 (12) |
| Physician | 1,406 (59) | 1,273 (59) | 1,282 (65) |
| Other | 13 (< 1) | 11 (1) | 9 (< 1) |
| **Limited to Physicians** | **N = 1,406** | **N = 1,273[g]** | **N = 1,282** |
| Specialty, n (%) | | | |
| Primary care | 590 (42) | 580 (46) | 532 (42) |
| Medical specialty | 554 (39) | 469 (37) | 524 (41) |

*(Continued)*

**Table 1.** (Continued)

| Patient Characteristics | Total (N = 348,739[a]) | Any In-Person (N = 313,980[b]) | Any Not-In-Person (N = 82,219) |
|---|---|---|---|
| Surgical specialty | 170 (12) | 145 (11) | 155 (12) |
| Other[h] | 92 (7) | 79 (6) | 71 (6) |
| Years since medical school graduation[i], median (IQR) | 21 (14, 30) | 21 (13, 30) | 21 (14, 30) |
| Ambulatory Clinical Full Time Equivalent, n (%) | | | |
| 0–0.25 | 626 (45) | 528 (41) | 522 (41) |
| 0.26–0.5 | 371 (26) | 345 (27) | 359 (28) |
| 0.51–0.75 | 232 (17) | 225 (18) | 225 (18) |
| 0.75–1 | 177 (13) | 175 (14) | 176 (14) |
| Prescribing volume, median prescriptions during study period (IQR) | 282 (130, 563) | 309 (145, 637) | 306 (145, 637) |
| Proportion of all prescriptions accounted for by antibiotics, Median % (IQR) | 6 (3, 11) | 6 (3, 11) | 6 (3, 10) |

a. Percentages may not add up to 100% because of rounding.

b. There were 47,460 patients (14% of all patients) in both the "any in-person" and "any not-in-person" groups. See Methods for the rationale for not performing statistical testing between groups.

c. 20 patients with missing data for gender.

d. There were 1743 clinicians (73% of all clinicians) in both the "any in-person" and "any not-in-person" groups.

e. 5 clinicians with missing data for gender.

f. APN is Advanced Practice Nurse. NP is Nurse Practitioner.

g. There were 1149 physicians (82% of all physicians) in both the "any in-person" and "any not-in-person" groups.

h. The three most common "Other" specialties were ophthalmology (52%), psychiatry (10%), and podiatry (9%). Northwestern Medicine does not provide dental care.

i. 203 physicians with missing data for years since medical school graduation

**Table 2. Antibiotic prescriptions by encounter type and diagnosis category (N = 714,057).**

| | In-Person | Not-In-Person | | | | | | OverallRow % |
|---|---|---|---|---|---|---|---|---|
| | | Phone | Orders only | Refill | Patient Portal | Other[a] | All Not-In-Person | |
| | | %[b] | | | | | | |
| Chronic[c] | 15 | 19 | 17 | 29 | 22 | 16 | 20 | 16 |
| Appropriate or potentially appropriate | 59 | 31 | 37 | 20 | 26 | 27 | 30 | 54 |
| *Appropriate*[d] | *16* | *11* | *15* | *6* | *7* | *9* | *11* | *15* |
| *Potentially appropriate*[e] | *43* | *20* | *22* | *14* | *19* | *17* | *20* | *39* |
| Non-antibiotic-appropriate[f] | 25 | 7 | 9 | 6 | 6 | 28 | 8 | 22 |
| No diagnosis | <1 | 44 | 37 | 45 | 47 | 30 | 42 | 8 |
| Overall Column % | 81 | 10 | 5 | 3 | 2 | <1 | 19 | 100 |

a. The 5 most common "Other" encounter types were Clinical Support, EpicOnHand Encounter (a miscellaneous type), Outpatient Testing, Pharmacy Consult, and Letter (Out).

b. Percentages may not add up to 100% because of rounding. Column percentages, except for the last row. N = 578,130 (81%) in-person prescriptions and 135,927 (19%) not in-person prescriptions.

c. The 5 most common chronic diagnoses within the 6 months prior to prescribing were chronic sinusitis, unspecified; acne vulgaris; chronic obstructive pulmonary disease, unspecified; rosacea, unspecified; and chronic rhinitis.

d.The 5 most common same-day antibiotic-appropriate diagnoses were urinary tract infection, site not specified; streptococcal pharyngitis; acute cystitis without hematuria; acute cystitis with hematuria; and pneumonia, unspecified organism.

e. The 5 most common same-day potentially appropriate infection-related diagnoses were acute sinusitis, unspecified; acute pharyngitis, unspecified; acute maxillary sinusitis, unspecified; acute pansinusitis, unspecified; and acute frontal sinusitis, unspecified.

f. The 5 most common same-day non-antibiotic-appropriate diagnoses were acute upper respiratory infection, unspecified; acute bronchitis, unspecified organism; bronchitis; cough; and encounter for immunization.

Not-in-person encounters were more likely associated with no diagnosis code (42% versus <1%). Said another way, 58% of not-in-person encounters had a diagnosis whereas 99% of in-person encounters had a diagnosis.

## Discussion

In this study of four years of antibiotic prescribing data within an integrated academic health system, we found that 19% of ambulatory antibiotics were prescribed with not-in-person encounters. Antibiotics prescribed without in-person visits were more likely for chronic diagnoses; less likely to be associated with appropriate or potentially appropriate diagnoses; and more likely to be associated with no diagnosis. These findings suggest that ambulatory stewardship interventions that focus only on in-person visits may miss a large proportion of antibiotic prescribing, inappropriate prescribing, and antibiotics prescribed in the absence of any diagnosis. The absence of a diagnosis may say more about inappropriate documentation than inappropriate antibiotic prescribing. However, if one considered the absence of a diagnosis to be inappropriate prescribing, 25% of in-person antibiotics were inappropriate and 50% of not-in-person antibiotics were inappropriate.

Our findings are particularly important considering the COVID-19 pandemic. During the pandemic, there were shifts in visit numbers: in-person visits have become less frequent and use of not-in-person encounters have become more common [27]. At the study health system, for example, there are now many more synchronous telephone and video visits that require a billing code. There were also pandemic shifts in antibiotic prescribing [28–30].

Familiarity with remote care and improved reimbursement for synchronous, remote visits has the potential to improve all-ambulatory-antibiotic stewardship. Alternatively, increased remote care has the potential to be associated with decreased antibiotic prescribing quality. Analyses of all-antibiotic prescribing, appropriateness, and visit type should be repeated and the results validated in the COVID pandemic and post-pandemic periods. Given the shifts in visit types and antibiotic prescribing, a continued focus only on antibiotic prescribing during in-person visits or solely on telehealth visits is less and less justifiable. Similarly, studies that only focus on telemedicine may only capture a fraction of overall antibiotic prescribing at a health system [31–33].

Our appropriateness findings are similar to what Fleming-Dutra and colleagues reported using nationally representative survey data from the United States. After sampling from ambulatory visits with oral antibiotic prescriptions they found an estimated annual antibiotic prescription rate of 506 per 1000 population and an estimated 353 (70%) were likely clinically appropriate [34]. We have conducted studies using claims data and found between 26% and 29% were not associated with a recent diagnosis code [5, 21], quite a bit higher than the 8% from the present study. Regarding not-in-person encounters, we previously found that about 30% of antibiotic fills were "non-visit-based" [22, 23] which is more than the 19% of not-in-person encounters in the present study. A key difference is that in the prior studies we had a more restrictive definition of chronic antibiotic prescribing, based on long-term dispensing. In the present study, to be more lenient to clinicians, we initially associated antibiotic prescriptions with chronic conditions based on 6 months of diagnoses. Claims and EHR data might differentially detect antibiotics: antibiotics might be prescribed in the EHR, but not generate a claim (i.e., the patient paid cash or did not pick up the antibiotic) or clinicians could circumvent EHR prescribing by calling a pharmacy directly. Compared to claims data, EHR data may be more likely to demonstrate the relationships among prescribers, antibiotics, sites of care, and diagnoses. Claims data may rely on different information sources for antibiotic claims, prescriber information, and diagnosis codes and overstate the "inappropriateness" of antibiotic prescribing relative to EHR data.

Our study is subject to several limitations beyond it being conducted prior to the pandemic. First, it was conducted within a single integrated health delivery system which may limit generalizability. Second, the data we analyzed were dependent on clinician diagnosis code selection within the EHR which is sometimes an incomplete reflection of actual care. Third, we were limited to prescribing data (e.g., clinician behavior) and did not have data regarding if or when patients filled and took their prescribed antibiotic (e.g., patient behavior).

Fourth, in determining appropriateness, we erred on the side of leniency. For example, we considered all antibiotic prescribing for chronic conditions to be acceptable (e.g., emphysema, chronic sinusitis, and chronic rhinitis) even though the prescribing may not have been clinically appropriate. We also forgivingly assumed that the most antibiotic-appropriate diagnosis was the reason for the antibiotic prescription, did not require the antibiotic class to be appropriate for a given condition, and could use a single diagnosis code to justify the appropriateness of multiple antibiotics. Thus, our study's estimate of inappropriate prescribing is undoubtedly a lower bound.

Fifth, our approach assumed that antibiotic prescriptions on different days resulted from a separate episode of care, but this may not always be the case. For example, if a follow-up telephone visit by a patient or pharmacist to clarify an antibiotic written at an in-person visit resulted in a new antibiotic prescription, this would have been counted as two separate prescriptions. However, any resulting bias is likely minimal, as only 2% of antibiotic prescriptions had another antibiotic prescription in the subsequent 1 to 3 days. Sixth, it was beyond the scope of the current study to examine prescriber-level variability in antibiotic appropriateness, including whether certain prescribers disproportionately accounted for not-in-person antibiotic prescribing that lacked diagnosis codes or whether prescribers had different patterns of appropriateness across their in-person and not-in-person encounters. Seventh, direct comparisons of appropriateness between prescriptions written with in-person visits versus not-in-person encounters were limited by the frequent lack of diagnosis codes among prescriptions at not-in-person encounters. However, the lack of an associated diagnosis code could itself be viewed as inappropriate or a quality problem that would limit the effectiveness of ambulatory stewardship programs. Additional research is needed to examine the rationale of antibiotics prescribing not associated with a diagnosis code.

In summary, to decrease unnecessary antibiotic prescribing, improve patient safety, and slow the development of antibiotic-resistant bacteria, ambulatory antibiotic stewardship interventions should comprehensively measure all antibiotic prescribing, not just prescribing at in-person visits. The importance of considering prescribing across all encounters may only grow if ongoing shifts towards remote care continue.

## Supporting information

**S1 File. List of all included oral antibiotic prescriptions.**
(PDF)

**S1 Data. Classification of all 94,249 International Classification of Diseases, 10$^{th}$ Edition, Clinical Modification diagnosis codes as chronic, antibiotic-appropriate, potentially antibiotic-appropriate, and non-antibiotic-appropriate.**
(XLSX)

## Author Contributions

**Conceptualization:** Tiffany Brown, Michael A. Fischer, Mark W. Friedberg, Kao-Ping Chua, Jeffrey A. Linder.

**Data curation:** Tiffany Brown, Ji Young Lee, Adriana Guzman, Kao-Ping Chua, Jeffrey A. Linder.

**Formal analysis:** Tiffany Brown, Ji Young Lee, Michael A. Fischer, Kao-Ping Chua, Jeffrey A. Linder.

**Funding acquisition:** Michael A. Fischer, Mark W. Friedberg, Jeffrey A. Linder.

**Investigation:** Adriana Guzman, Michael A. Fischer, Mark W. Friedberg, Kao-Ping Chua, Jeffrey A. Linder.

**Methodology:** Ji Young Lee, Michael A. Fischer, Mark W. Friedberg, Kao-Ping Chua, Jeffrey A. Linder.

**Project administration:** Tiffany Brown, Adriana Guzman, Jeffrey A. Linder.

**Resources:** Jeffrey A. Linder.

**Supervision:** Tiffany Brown, Michael A. Fischer, Mark W. Friedberg, Kao-Ping Chua, Jeffrey A. Linder.

**Validation:** Tiffany Brown, Ji Young Lee, Adriana Guzman, Michael A. Fischer, Mark W. Friedberg, Jeffrey A. Linder.

**Writing – original draft:** Tiffany Brown, Ji Young Lee, Adriana Guzman, Michael A. Fischer, Mark W. Friedberg, Kao-Ping Chua, Jeffrey A. Linder.

**Writing – review & editing:** Tiffany Brown, Ji Young Lee, Adriana Guzman, Michael A. Fischer, Mark W. Friedberg, Kao-Ping Chua, Jeffrey A. Linder.

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
