## [Decision Letter · Decision Letter 0]

6 Mar 2023

PONE-D-23-01039Prevalence and Appropriateness of In-Person versus Not-In-Person Ambulatory Antibiotic Prescribing in An Integrated Academic Health System: A Cohort StudyPLOS ONE

Dear Dr. Linder,

Thank you for submitting your manuscript to PLOS ONE. After careful consideration, we feel that it has merit but does not fully meet PLOS ONE’s publication criteria as it currently stands. Therefore, we invite you to submit a revised version of the manuscript that addresses the points raised during the review process.

We look forward to receiving your revised manuscript.

Kind regards,

Juan F. Orueta, MD, PhD

Academic Editor

PLOS ONE

Journal Requirements:

   "This research was supported by grant number R01HS024930 from the Agency for Healthcare Research and Quality (AHRQ). The contents of this product are solely the responsibility of the authors and do not necessarily represent the official views of or imply endorsement by AHRQ or the U.S. Department of Health and Human Services. The funders had no role in study design, data collection or analysis, decision to publish, or preparation of the manuscript"

  "This research was supported by grant number R01HS024930 from the Agency for Healthcare Research and Quality (AHRQ). The contents of this product are solely the responsibility of the authors and do not necessarily represent the official views of or imply endorsement by AHRQ or the U.S. Department of Health and Human Services. The funders had no role in study design, data collection or analysis, decision to publish, or preparation of the manuscript."

Reviewers' comments:

Reviewer's Responses to Questions

**Comments to the Author**

1. Is the manuscript technically sound, and do the data support the conclusions?

Reviewer #1: Yes

Reviewer #2: Yes

2. Has the statistical analysis been performed appropriately and rigorously? 

Reviewer #1: Yes

Reviewer #2: No

3. Have the authors made all data underlying the findings in their manuscript fully available?

Reviewer #1: No

Reviewer #2: Yes

4. Is the manuscript presented in an intelligible fashion and written in standard English?

Reviewer #1: Yes

Reviewer #2: Yes

5. Review Comments to the Author

Reviewer #1: Thank you for the chance to review this manuscript, which examines antibiotics prescribed in outpatient settings in an integrated health delivery system in the US, detailing those prescribed with in-person visits versus outside of in-person visit. They found that 19% of antibiotics were prescribed outside of in-person visits (primarily telephone visits). Compared to antibiotics prescribed with an in-person visit, those prescribed outside of an in-person visit were more likely to be chronic or to lack an associated diagnosis. Specific comments are below:

Major:

- I appreciate that diagnoses within a set period were extended across time and visits (all encounter diagnoses on the day of diagnosis by any provider and diagnoses by same provider across next 21 days being examined for possible antibiotic appropriate diagnoses). In contrast, it appears that categorization of visit type (in-person vs non-in-person) was more rigid. For example, if a pharmacy called to clarify a prescription on the day of or day after an in-person visit prompting a telephone encounter with a new prescription, was this classified as a telephone encounter or associated with the preceding in-person encounter? Were these counted as two separate antibiotic prescriptions in two different settings, or only ultimately counted as a telephone encounter (with the prior in-person prescription perhaps cancelled in the EHR?)? Sensitivity analysis around these assumptions seem important to understand out-of-visit prescribing.

- It seems important to re-emphasize that during the study period “at least one diagnosis was required for an in-person encounter, but not for a telephone or patient portal encounter.” Thus I would hesitate to interpret the 42% not-in-person visits without a diagnosis as inappropriate prescribing, but rather inappropriate documentation enabled by different requirements within the EHR for different visit types, leaving the question of appropriateness wide open for these visits, and necessitating further examination in a EHR where/when diagnoses are required.

Minor:

- Given the inclusion of chronic antibiotic use as a category of interest, unclear why antibiotics for urinary tract infection prophylaxis were specifically excluded.

- Would emphasize in methods and discussion/limitation that the specific antibiotic could be inappropriate for the specific diagnosis and yet still classified as “appropriate” in this schema.

- Are dental prescriptions included in these data? Given the focus on global ambulatory prescribing, would be worth noting whether this setting is included or not.

Thank you again for the opportunity to review.

Reviewer #2: This is a very interesting topic, although clearly the world has changed since the study's end. Overall, I had only one major issue.

The introduction is well reasoned.

Methods:

I would clarify you didn't look at antibiotics only prescribed for prophylaxis such as methenamine; nitrofurantoin and bactrim may be much more commonly used for UTI prophy. Also, dapsone for example?

I would assume that even post-COVID, patient portal encounters and many non-visit phone encounters do not require a billing code

The methods are well-described. However, was there a reason specifically for not doing testing for statistical significance? I understand and appreciate the clinical significance. It just makes it quite difficult to say, "more likely to be..."

The rate of patients receiving antibiotics for chronic conditions seems extremely high. Almost as high, in fact, as the total proportion of antibiotics prescribed outside of clinic visits. Although I'm not sure of research supporting this, it is certainly higher than in many large health systems. It is also odd that the most common chronic diagnoses include chronic sinusitis, COPD, and chronic rhinitis (and not, for example, trimethoprim-sulfamethoxazole for PJP prophylaxis). Chronic sinusitis and chronic rhinitis do not seem like conditions necessitating chronic antibiotics. COPD also seems like a very small minority of patients would require chronic antibiotics, as general use of chronic antibiotics goes against international guidelines in this condition. Should this just be "sometimes" rather than "chronic?" As these categories are mutually exclusive, the chronic condition results are somewhat misleading. The fact that not many of these antibiotics are prescribed over refills also makes me suspect that these antibiotics are not really being prescribed for chronic infections.

Discussion: I would expand on some of these limitations in the discussion as well.

6. PLOS authors have the option to publish the peer review history of their article (what does this mean?). If published, this will include your full peer review and any attached files.

Reviewer #1: No

Reviewer #2: No

---

## [Author Response · Author response to Decision Letter 0]

4 Apr 2023

Please see response to reviewers letter for all responses.

---

## [Decision Letter · Decision Letter 1]

22 Jun 2023

PONE-D-23-01039R1Prevalence and Appropriateness of In-Person versus Not-In-Person Ambulatory Antibiotic Prescribing in An Integrated Academic Health System: A Cohort StudyPLOS ONE

Dear Dr. Linder,

Thank you for submitting your manuscript to PLOS ONE. After careful consideration, we feel that it has merit but does not fully meet PLOS ONE’s publication criteria as it currently stands. Therefore, we invite you to submit a revised version of the manuscript that addresses the points raised during the review process.

The authors have provided adequate responses to the previous comments and sufficiently explained the limitations of the study. However, a minor point observed by the reviewer needs to be clarified.

We look forward to receiving your revised manuscript.

Kind regards,

Juan F. Orueta, MD, PhD

Academic Editor

PLOS ONE

Journal Requirements:

Reviewers' comments:

Reviewer's Responses to Questions

**Comments to the Author**

1. If the authors have adequately addressed your comments raised in a previous round of review and you feel that this manuscript is now acceptable for publication, you may indicate that here to bypass the “Comments to the Author” section, enter your conflict of interest statement in the “Confidential to Editor” section, and submit your "Accept" recommendation.

Reviewer #2: All comments have been addressed

2. Is the manuscript technically sound, and do the data support the conclusions?

Reviewer #2: Yes

3. Has the statistical analysis been performed appropriately and rigorously? 

Reviewer #2: Yes

4. Have the authors made all data underlying the findings in their manuscript fully available?

Reviewer #2: Yes

5. Is the manuscript presented in an intelligible fashion and written in standard English?

Reviewer #2: Yes

6. Review Comments to the Author

Reviewer #2: Thank you for addressing these comments. The article is now clearer. However, I have the most minor of revisions to request. On reviewing again, I am realizing that the distinction between adult vs pediatric patients (and internal medicine vs family medicine vs pediatrics) is unclear. Were pediatric patients included as well as adults?

7. PLOS authors have the option to publish the peer review history of their article (what does this mean?). If published, this will include your full peer review and any attached files.

Reviewer #2: **Yes: **Sara C Keller

---

## [Author Response · Author response to Decision Letter 1]

7 Jul 2023

Please see attached Response to Reviewers file.

---

## [Editor Report · Decision Letter 2]

17 Jul 2023

Prevalence and Appropriateness of In-Person versus Not-In-Person Ambulatory Antibiotic Prescribing in An Integrated Academic Health System: A Cohort Study

PONE-D-23-01039R2

Dear Dr. Linder,

We’re pleased to inform you that your manuscript has been judged scientifically suitable for publication and will be formally accepted for publication once it meets all outstanding technical requirements.

Kind regards,

Juan F. Orueta, MD, PhD

Academic Editor

PLOS ONE

---

## [Editor Report · Acceptance letter]

18 Jul 2023

PONE-D-23-01039R2 

Prevalence and Appropriateness of In-Person versus Not-In-Person Ambulatory Antibiotic Prescribing in An Integrated Academic Health System: A Cohort Study 

Dear Dr. Linder:

I'm pleased to inform you that your manuscript has been deemed suitable for publication in PLOS ONE. Congratulations! Your manuscript is now with our production department. 

Kind regards, 

on behalf of

Dr. Juan F. Orueta 

Academic Editor

PLOS ONE